# Hyaluronate Functionalized Multi-Wall Carbon Nanotubes Filled with Carboplatin as a Novel Drug Nanocarrier against Murine Lung Cancer Cells

**DOI:** 10.3390/nano9111572

**Published:** 2019-11-06

**Authors:** Daniel Salas-Treviño, Odila Saucedo-Cárdenas, María de Jesús Loera-Arias, Humberto Rodríguez-Rocha, Aracely García-García, Roberto Montes-de-Oca-Luna, Edgar I. Piña-Mendoza, Flavio F. Contreras-Torres, Gerardo García-Rivas, Adolfo Soto-Domínguez

**Affiliations:** 1Departamento de Histología, Facultad de Medicina, Universidad Autónoma de Nuevo León, Monterrey C.P. 64460, Mexico; qcbdanielsalas@outlook.es (D.S.-T.); odilam@hotmail.com (O.S.-C.); marichu_loera@yahoo.com.mx (M.d.J.L.-A.); humbertordz54@gmail.com (H.R.-R.); Aracely_79_20@yahoo.com (A.G.-G.); rrrmontes@yahoo.com (R.M.-d.-O.-L.); edgar.pinamen@gmail.com (E.I.P.-M.); 2Departamento de Genética Molecular, Centro de Investigación Biomédica del Noreste (CIBIN) del IMSS, Monterrey C.P. 64720, Mexico; 3Escuela de Ingeniería y Ciencias, Tecnológico de Monterrey, Monterrey C.P. 64849, Mexico; contreras.flavio@tec.mx; 4Escuela de Medicina, Tecnológico de Monterrey, Monterrey 64849, Mexico; gdejesus@tec.mx

**Keywords:** multi-wall carbon nanotubes, nanomedicine, tumor cells, hyaluronate, carboplatin

## Abstract

Carbon nanotubes (CNTs) have emerged in recent years as a potential option for drug delivery, due to their high functionalization capacity. Biocompatibility and selectivity using tissue-specific biomolecules can optimize the specificity, pharmacokinetics and stability of the drug. In this study, we design, develop and characterize a drug nanovector (oxCNTs-HA-CPT) conjugating oxidated multi-wall carbon nanotubes (oxCNTs) with hyaluronate (HA) and carboplatin (CPT) as a treatment in a lung cancer model in vitro. Subsequently, we exposed TC–1 and NIH/3T3 cell lines to the nanovectors and measured cell uptake, cell viability, and oxidative stress induction. The characterization of oxCNTs-HA-CPT reveals that on their surface, they have HA. On the other hand, oxCNTs-HA-CPT were endocytosed in greater proportion by tumor cells than by fibroblasts, and likewise, the cytotoxic effect was significantly higher in tumor cells. These results show the therapeutic potential that nanovectors possess; however, future studies should be carried out to determine the death pathways involved, as well as their effect on in vivo models.

## 1. Introduction

Carbon nanotubes (CNTs) are cylindrical nanostructures first described in 1991 [1]. CNTs are a carbon allotrope, and their structure can be elucidated as a rolled sheet of graphene, which is formed for hexagonally bound sp2 carbon atoms [2]. Based on the number of graphene sheets found their structure can be classified as single-walled carbon nanotubes (SWCNTs) or multiple wall carbon nanotubes (MWCNTs) [3]. CNTs diameter can range from 0.4 to 2 nm for SWCNTs and 10 to 100 nm for MWCNTs, while the length could measure a few microns (10 to 50 μm) [4]. CNTs have special features like hollow structure, electrical conductivity, and high aspect ratio that make them strong candidates for use in nanomedicine [5], especially for drug delivery, biosensors or tissue scaffolds [6]. However, previous studies described that cell interaction with CNTs produces acute inflammation, apoptosis, the formation of reactive oxygen species (ROS) and death induction by autophagy [7,8].

Functionalization in CNTs is a very successful process to increase biocompatibility and dispersibility in aqueous solutions, this is due to their high aspect ratio, allowing to couple a greater number of molecules on the surface [9]. Functionalization promotes interaction of CNTs with biomolecules [7], prevents the formation of toxic aggregates [3] and makes them suitable for drug delivery or proteins and nucleic acid cleavage on inner or outer sheets [10], e.g., hyaluronic acid (HA). This compound act as a ligand of the CD44 receptor, which is overexpressed in a large number of tumors and has been widely used to biodirect MWCNTs for drug delivery [11,12,13]. In this sense, MWCNTs are more resistant to functionalization than SWCNTs, since this process involves the breakage of C = C bonds in graphene, which would leave it exposed to breaks in the latter [14].

Since the discovery of the potential use of CNTs as drug carriers, numerous studies have been conducted to demonstrate the efficacy and safety of such nanomaterials, using a variety of antineoplastic molecules like doxorubicin [15], gemcitabine [4], paclitaxel [14], and diverse platinum-based drugs [16].

Carboplatin (CPT) is an FDA approved homologous drug of cisplatin, with differences in its pharmacokinetics and pharmacodynamics very frequently used as chemotherapy for lung cancer and various types of cancers [17]. Previous in vitro studies have analyzed the antineoplastic efficacy of CPT coupled to CNTs, finding that breast, prostate, kidney, and bladder tumor cell lines are more sensitive to CNTs–CPT nanostructures than CPT alone, probably due to the drug protection and prolonged release effect conferred by the CNTs [18,19].However, the use of functionalized CNTs to carry platinum-based drugs to optimize its delivery in a lung cancer model has never been attempted.

Although HA has been widely used to biodirect MWCNTs for drug delivery, in our study, we constructed a nanovector based on CNTs functionalized with HA to directed delivery of CPT in a cellular model of murine lung cancer, using mouse TC-1 cells as a tumor line and NIH/3T3 as a non-tumor cell line. We performed cytotoxic characterization and selectivity tests to evaluate the efficacy and safety of our nanocomposite.

## 2. Materials and Methods

### 2.1. Nanovectors Construction

The MWCNTs were purchased from Nanostructured and Amorphous Materials, Inc. (Houston, TX, USA). The graphitized MWCNTs have a 99.9% purity (stock: #1240YJF) and are slightly functionalized with carboxyl groups (-COOH) that represent less than 0.26% by weight. MWCNTs diameter range was 50 to 80 nm with an internal diameter of 5 to 15 nm, while the length is between 10 to 20 μm.

First, the commercial carbon nanotubes (cCNTs) were oxidized with acid treatment to increase the number of carboxyl groups on the surface. For this, 40 mg of cCNTs were added to a solution of 8 mL of 3:1 ratio of H_2_SO_4_: HNO_3_, this mixture was ultrasonicated for 15 min and homogenized under vigorous magnetic stirring for 5 h. Subsequently, the oxidized carbon nanotubes (oxCNTs) were separated by centrifugation and washed to a neutral pH and finally resuspended in acetone and dried in an oven at 60 °C. oxCNTs functionalized with HA (oxCNTs–HA) were prepared according to previous reports [20]. Briefly, 20 mg of oxCNTs were dispersed in 8 mL of DMSO by ultrasonication for 30 min and activated with a solution of EDC-HCl (N-(3-Dimethylaminopropyl)-N′-ethyl carbodiimide hydrochloride) and NHS (N-Hydroxysuccinimide) in DMSO (20 mg and 10 mg respectively in 0.8 mL) homogenized with magnetic stirring, allowing activation of carboxyl groups for 3 h in magnetic stirring and the solution of polyethyleneimine (20 mg in 2 mL of DMSO) as coupler is added, allowing to react for 48 h, then separated by centrifugation and left in the oven at 60 °C until dry. Subsequently, HA was solubilized in DMSO (20 mg in 1.5 mL) and activated with a solution of EDC-HCl in DMSO (20 mg in 0.8 mL) for a period of 3 h. Finally, the activated HA solution was added dropwise to the oxCNTs solution; then, this reaction was homogenized for 48 h to create oxCNTs–HA nanovectors. Finally, CPT was incorporated into the inner space of the oxCNTs–HA through the nano-extraction method described previously [21]. Briefly, 5 mg of oxCNTs–HA were resuspended in 5 mL of pure ethanol, and 5 mg of CPT were added. Then, the solution was allowed homogenizing by magnetic stirring for 72 h. Finally, the oxCNTs–HA–CPT were washed with methanol 3 times and dried in an oven at 60 °C for 48 h.

### 2.2. Characterization of the Constructed CNTs Using Transmission Electron Microscopy (TEM), Infrared Spectroscopy (FTIR), Thermogravimetric Analysis (TGA) and Hydrophobicity Index Test

The morphologic characteristics (size and thickness) of the CNTs were analyzed by TEM with a Carl Zeiss EM-109 microscope at 80 kV, the samples were prepared by dispersing the CNTs in DMSO (100 μg/mL) in an ultrasound bath for 30 min and 10μLof the solution was deposited on carbon-coated copper grid and dried at room temperature, observations were made at 20,000×.

To look for experimental evidence of chemical functionalization in CNTs with HA through the covalent link, Fourier Transform Infrared (FTIR) spectroscopy was used. FTIR spectra were recorded on a Spectrum One FTIR spectrophotometer (PerkinElmer, Shelton, CT, USA) using the attenuated total reflectance (ATR) mode at room temperature (RT). cCNTs, oxCNTs, and oxCNTs-HA were deposited as dry powders onto a zinc selenide (ZnSe) window located in a MIRacle ATR accessory. The powders were previously dispersed from mechanical exfoliation in agate mortar; no solvent was used through this process. In addition, the amount of grafted HA onto oxCNTs was determined by thermogravimetric analysis (TGA) and differential scanning calorimetry (DSC). TGA and DSC curves were acquired by using an SDT Q600 thermogravimetric analyzer (TA Instruments). Approximately 20 mg of powdered sample was heated in an open aluminum pan (90μL) under an N_2_ (99.999%) flow of 100 mL·min^−1^ with a heating ramp of 10 °C·min^−1^ until 1000 °C. The curves recorded showed weight loss (mg).

The hydrophobicity index test is used to know the degree of solubility that a certain compound presents to polar and non-polar solvents. For this, we used a previously reported method [22], briefly, solutions of CNTs were made in a range of concentrations from 10 to 100 μg/ml in 3 mL of H_2_Od and were sonicated for 15 min, and an aliquot was taken to measure its absorbance at 550 nm (A_0_). Subsequently, 3 mL of 1-octanol were added to these solutions, then mixed by vortex for 15 **s** and left to stand for 30 min at room temperature, after this time, an aliquot of the aqueous phase was taken and read at 550 nm (A_1_), and the hydrophobicity index was calculated according to the following equation:(1)$$\left[\frac{A_{0}\;-{\;A}_{1}}{A_{0}}\right]\;\times 100\%,$$

### 2.3. Cell Culture

In this study, we used two types of cells: The tumor cell line TC-1 (mouse pulmonary epithelium transformed with HPV-16 E6 and E7 proteins) and the non-tumor cell line NIH/3T3 (rat fibroblast), both were obtained from the ATCC (Manassas, VA, USA). TC-1 cells were cultured using RPMI 1640 supplemented with 10% inactivated fetal bovine serum (FBS), while DMEM supplemented with 10% newborn calf serum was used for NIH/3T3 cells. Also, the HeLa cervical cancer cell line was used as a positive control of the immunohistochemistry (IHC) to detect CD44; and these were cultured with DMEM medium supplemented with 10% FBS. All cell lines were kept in an incubator at 37 °C in an atmosphere containing 5% CO_2_. For the cell capture, viability and ROS assays, the cells were treated with the different types of CNTs, which were resuspended in the corresponding culture media for each cell line (supplemented with FBS).

### 2.4. Evaluation of Cell Uptake of the Nanovectors in the Tumor and Non-Tumor Cell Lines by Light Microscopy and TEM

Since the CNTs enter the cells by means of phagocytosis, forming intracytoplasmic vesicles, it is necessary to embed the cells with epoxy resins to obtain ultrathin sections to be analyzed by TEM and semi-thin sections for bright field microscopy. For this, we incubated 1 × 10^6^ cells in 60 mm dishes and allowed them to adhere for 24 h. Subsequently, they were treated with cCNTs, oxCNTs, oxCNTs-HA-CPT and CPT at 10 μg/ml for 24 h in culture medium, were harvested using 0.25% trypsin, washed twice with 1× sterile PBS (pH: 7.4) and they were fixed with 2.5% glutaraldehyde for 24 h and post-fixation with 2% OsO_4_ for 30 min. The samples were dehydrated with an acetone gradient and embedded in epoxy resin (Resin embed-812, EMS. # Cat: 14120). Finally, semi-thin (150 μm thick, toluidine blue-stained) and thin (80 μm thick, lead citrate and uranyl acetate contrasted) sections were obtained by ultramicrotomy.

In semi-thin sections, cells with endoplasmic vesicles containing nanovectors were counted, considering a total of 300 cells per histological preparation (*n* = 5) to yield the percentage of positive cells in each treatment. Moreover, in ultrathin sections, the endoplasmic vesicles were observed and described.

### 2.5. MTT Assay

The mitochondrial function of the cell can be evaluated based on the activity of the reductases found in this organelle. To determine such activity, we decided to use the MTT assay [3-(4,5-dimethylthiazol-2-yl)-2,5-diphenyltetrazolium bromide], which is reduced to formazan by such enzymes. 7.5 × 10^3^ cells per well (*n*=7) were incubated in a 96-well plate and allowed to adhere for 24 h. Subsequently, CNTs and CPT were added to cells for 24 h. After that, MTT was added to each well and incubated for 3.5 h at 37 °C, then MTT was removed, and MTT solvent (4mM HCl, 0.01% NP40 in isopropanol) was added, the plate was stirred for 15 min before absorbance reading at 590 nm.

### 2.6. Determination of ROS

Reactive oxygen species (ROS) were determined using the dihydroethidium (DHE) probe, which is a molecule that oxidizes in the presence of oxidizing radicals such as the superoxide ion (O_2_^−^) or hydrogen peroxide (H_2_O_2_) generates fluorescence. In this study, ROS were measured by flow cytometry, based on a method previously established by our research group [23]. Briefly, 1 × 10^6^ cells were plated on 60 mm plates and left 24 h for adhesion. Subsequently, the cells were treated with CNTs and CPT for 24 h and then exposed to DHE (10 μM) for 15 min, for the negative control (basal oxidative stress), the cells had only a renewal of fresh medium supplemented when administered treatments. Then, the cells were harvested, washed with 1×PBS and resuspended to be measured by flow cytometry (Muse Cell Analyzer, Millipore Sigma).

### 2.7. Statistic Analysis

Descriptive Statistics: Qualitative variables were summarized by calculating absolute and relative frequencies in percentage and Confidence Intervals (CI) at 95%. The quantitative variables were summarized by means of the calculation of measures of mean and/or median central tendency and, measures of dispersion, standard deviation (SD) and/or distance of the variable. For the calculation of the CI, the typical error of the mean was obtained.

Inferential Statistics: In the possible groups defined by the modalities of qualitative variables in the sample, the Chi-Square test (yacht correction) was applied. For 2 × 2 contingency tables, Fisher’s exact test was applied. The calculations were made with the SPSS v17 program. In all the analyzes, the criteria of α ≤ 0.05, 1–β = 80% and two-tailed tests were applied.

## 3. Results

### 3.1. Morphological and Chemical Characterization of Constructed Nanovectors

The nanovectors obtained commercially, and those produced in this study were first analyzed through TEM. The oxCNTs – HA – CPT presented better dispersion and have thicker and dense walls compared to those commercially acquired (Figure 1a).

To elucidate the organic component of functionalized nanovectors, we perform FTIR spectroscopy and TGA. The ATR-FTIR spectra (baseline uncorrected) of cCNTs and ox–CNTs–HA samples are shown in Figure 1b. Spectral features in the functionalized CNTs appear at 1400–1800 cm^−1^. In this region we can observe the typical peak of oxCNTs-HA at about 1700 cm^−1^ which is characteristic of the C=O carboxyl amide I group in HA, confirming the successful modification. Furthermore, the peaks of oxCNTs-HA located at 1584 cm^−1^ and 1489 cm^−1^ are due to stretching and vibration of CH moieties of HA. None of these distinctive features appeared in the spectrum of cCNTs, evidencing that the functionalization of HA in CNTs using polyethyleneimine has been carried out successfully.

TGA curves (Figure 1c) showed a slight and uniform weight loss of about 2–3%, due to physisorbed water occurred at the temperatures up to 110 °C. The contribution of oxidized moieties covalently bound to cCNTs was estimated in a percentage of about 2–3% for oxCNTs, further weight loss after 440 °C that can be related to a gradual decomposition of the oxidized walls; the weight loss is about 1% indicating that the percentage of oxidation is no longer than 3%. In the case of oxCNTs–HA samples, TGA indicates a weight loss of about 10% related to the decomposition for both organic moieties from HA and CNTs. However, the total decomposition of grafted HA can be observed at 366 °C in which the abrupt weight loss is about 2.5%. Finally, a further stepwise for oxCNTs–HA occurs in the range from 375–570 °C, which can be related to the CNTs entity per se, due to a gradual thermal decomposition of the structure.

From DSC curves (Figure 1d), there is observed that the calorimetric events occurred in different ways for the three samples analyzed. These curves indicate all the exothermic processes evidencing the decomposition of all structures. A broad band in all the DSC curves in the temperature range from 200–400 °C indicates a large exothermic event related to the heat output. A broad peak appears at about 425 °C for oxCNTs, indicates a decomposition unlike the observed for cCNTs and supporting the fact that two samples are different in nature. More important, a broader peak at 600 °C is observed for oxCNT-HA indicating a unique exothermic event not observed in the other samples, and which demands a substantial amount of energy to decompose the sample. Therefore, the decomposition processes for the cCNTs, oxCNTs, and oxCNTs-HA samples proceeded in a noticeably different way that can be characterized by strong exothermic events occurring in the corresponding sample analyzed.

In addition, the hydrophobicity index test was significantly lower in the oxCNTs-HA (Figure 1e). As a result, the constructed nanovectors have a better dispersion and stability in aqueous media, compared to those acquired commercially.

### 3.2. Analysis of Constructed Nanovectors Uptake by TC-1 and NIH/3T3 Cell Lines

After the administration of the treatments with CNTs for 24 h to TC-1 and NIH/3T3 cells, we found different levels of uptake of nanovectors. The uptake of cCNTs (Figure 2a) in TC-1 cells was low, and as expected, oxCNTs-HA-CPT were taken up in a higher amount because of the high expression of the receptor for HA (Figure 2b). To validate these observations, we realize TEM analysis and we could find single vesicles in the cytoplasm of TC-1 cells treated with cCNTs, (Figure 2c) moreover, with the treatment of oxCNTs-HA-CPT (Figure 2d) multiple vesicles were observed and nanovector uptake was higher (Figure 2e).

### 3.3. Therapeutic Efficacy of Constructed Nanovectors in Tumor and Non-Tumor Cells

The cytotoxic efficacy of antineoplastic compounds is by far the most important characteristic because it depends on their kinetics, stability, and activity. To determine this property, we use the MTT assay, which evaluates the activity of mitochondrial enzymes. After 24 h of treatment with the cCNTs, the fibroblasts showed a decrease in the mitochondrial activity of 25% of normal levels regardless of the type of nanotube, and this effect was not dose-dependent; only in the CPT treatment, the activity was significantly lower in the highest concentration (Figure 3a). Moreover, tumor cells showed a decrease in metabolic activity in a dose-dependent manner when exposed to cCNTs and CPT; nevertheless, this effect was much more pronounced in the oxCNTs–HA–CPT, evidencing the optimization of the antitumor effect of this drug when it is introduced into functionalized CNTs and shows high selectivity towards tumor cells (Figure 3b).

### 3.4. Oxidative Stress Induction by Constructed Nanovectors in TC-1 and NIH/3T3 Cells

Reactive oxygen species (ROS) generate oxidative damage to various cellular targets such as DNA, proteins, and lipids, which alters cell metabolism and induces death pathways [24]. DHE is a molecule frequently used to quantify the oxidative state in cells and is specific for O_2_^−^ or H_2_O_2_ radicals. In general, TC-1 cells showed higher ROS levels than NIH/3T3 (Figure 3c). Interestingly, when cCNTs were administered, a significant decrease in oxidative status was observed in both the TC-1 and NIH/3T3 cells compared to the basal levels of each cell type. However, with oxCNTs–HA–CPT, only a slight increase in ROS was observed in the TC-1 cells.

## 4. Discussion

The aim of this study was to evaluate the uptake, cytotoxicity, and induction of oxidative stress of a nanocarrier made of hyaluronate—functionalized CNTs loaded with CPT in murine tumor and non-tumor cells. To date, there are few studies that have attempted to efficiently deliver CPT on in vitro cancer models and in those that exist, the advantage of functionalization in CNTs has not been used to direct these compounds [18,19,25].

Currently, many nanocomposites have been designed and developed to optimize diagnosis or drug delivery, as each one has different capacities and properties, it becomes essential to characterize these compounds morphologically and physicochemically. In our study, the nanotubes were treated with acids to oxidize them, thus, increasing their hydrophilicity, dispersibility, reducing their cytotoxicity and increasing the binding capacity to HA, which will provide specificity to be internalized preferentially by cancer cells [20,26,27].

In TEM analysis, we could observe that in the oxCNTs-HA, the thickness of the walls and the dispersion increase as in other previously published works [11]. However, due to the limitations of the microscope used, we were not able to identify the CPT signals within the nanotubes, which have been reported to be observed as electrodense dots in the inner walls [25]. In the FTIR, the HA-functionalized nanotubes showed distinctive bands at 1400–1800, especially the peak at 1702 cm^−1^, representative of the amidation of the carbonyl group at the position I of the HA [11].

To ensure and quantify the presence of organic compounds (HA) bound to the CNTs, TGA was used. This analysis showed that the oxCNTs-HA contained at least 2.5% w/w of HA that decomposes completely at 366 °C and this weight loss is not observed in the oxCNTs nor in cCNTs.

The amount of HA in the oxCNTs–HA was slightly lower compared to other reports, this may be due to differences in the functionalization process, other studies report the use of 2′2 (ethylenedioxy) bis (ethylene amine) [27] or add an acylation step before amination to obtain higher reaction yields [13]. Finally, we confirm the increase in dispersibility and stability in aqueous solutions of oxCNTs-HA, since they had a significantly lower hydrophobicity index [22].

It has been reported that the capture and internalization of CNTs in non-phagocytic cells occurs through transport mediated by clatrine, caveolins, and micropinocytosis [6], Now, it is known that tumor cells overexpress folate or HA receptors [20], thus, these molecules have been successfully used as targeting systems in CNTs to attack these cells to a greater extent than healthy tissue [20,27,28,29,30], However, to date, there is not any study using HA with CPT, a drug widely used and approved by the FDA for therapies against many types of cancer [17], We observed that the capture of our nanovector by tumor cells was significantly higher than in non-tumor cells, proving that it has a good selectivity towards neoplastic cells.

Once we show that the nanovectors were well constructed and that they are preferably internalized in tumor cells, we evaluated the cytotoxic activity they possess. Although the A549 cell line (human pulmonary adenocarcinoma) is widely used to test the cytotoxicity and safety of nanomaterials targeting the lung, we decided to apply our nanovectors first in an animal tumor model because we intend to perfect the ideal composition of the nanovector to scale it in an animal in vivo in the near future. For this purpose, we use the TC-1 tumor cell line, these cells are mouse pulmonary epithelial cells transformed with HPV-16 E6 and E7 proteins, and form metastasic tumors specifically in lungs when inoculated intravenously [31] In previous studies, it has been observed that MWCNTs administered intravenously, are retained and reside in lung tissue during the first 48 h, reaching a peak of maximum concentration at 24 h [32]. In addition, CPT is a drug widely used in chemotherapy for lung cancer [33]; therefore, we consider that this cell line has good characteristics to test our nanovector in the context of a tumor model. In the cytotoxicity tests, the maximum concentration we tested was 50 μg/ml for cCNTs, ox-CNTs-HA-CPT, and CPT because until that concentration, the cells showed good tolerance to empty cCNTs. The cytotoxic effect of the constructed nanovector showed to be at least twice compared to the CPT alone. Other research groups have also observed an improvement in the cytotoxicity of platinum drugs [19,21,25,34], doxorubicin [20,27] and paclitaxel [35] when they internalize or bind to CNTs.

It is well known that ROS are strongly related to DNA mutations and tumorigenesis [36], and that tumor cells are deficient in antioxidant enzymes [37], so it is predictable that TC-1 cells have higher baseline levels of oxidative stress than fibroblasts. Studies show that MWCNTs induce oxidative stress in vitro [38], however, this effect is nullified when CNTs are purified to remove traces of heavy metals used in their synthesis such as iron, zinc or cobalt [5]. In our study, we use highly purified CNTs with very low amounts of Fe (65 ppm) or Al (10 ppm), so when incubated with the cells they do not cause an increase in ROS, and only a slight increase in the case of ox- CNTs-HA-CPT in TC-1 cells and this effect can be attributed to the drug [39]. On the other hand, we observe that the ROS levels of the fibroblasts were reduced when treated with the CNTs, this may be possible because the CNTs have a great electronic affinity on their molecular orbitals of the carbon atoms of their graphene walls, acting as a free radical scavenger [40,41], in addition, we observe that this effect is enhanced by the addition of hyaluronic acid, since it has antioxidant activity as well [42]. The preliminary results of present study show that ox-CNTs-HA-CPT have a good and specific antitumor effect in vitro; however, further studies should be carried out to determine the cell death pathways involved, the amount of CPT in the nanovector, as well as its application in an in vivo tumor model, in order to know the effects of the nanovector in a broader sense.

## 5. Conclusions

In this research, we have demonstrated the therapeutic potential of a nanovector based on MWCNTs in vitro. The physicochemical properties such as the functionalization capacity presented by cCNTs represent a very useful tool to biodirect them towards tumors, reducing cell death in adjacent healthy tissue. For this purpose, our nanovector was coupled with HA, and although there are many studies that apply this molecule to biodirect nanoparticles, there are no reports in the present literature of the use of cCNTs with HA to carry platinum drugs. In addition, CPT, being an FDA-approved drug and widely used in chemotherapy for lung cancer, becomes a viable option to optimize its delivery. The results of our work indicate that the nanovector is easy to obtain, has adequate functionalization, good hydrophilicity, disperses well in culture medium and preferably attacks tumor cells and that they are internalized in a greater proportion in these, compared to normal cells. Although our results are promising so far, studies are needed to complete toxicity and safety profiles such as determining the amount of CPT loaded in nanotubes, long-term toxicity tests and the in vivo tumor model. These studies will help us to optimize the formulation and functionalization and increase the specific toxicity, the load of the drug and its cellular uptake.

## Figures and Tables

**Figure 1 nanomaterials-09-01572-f001:**
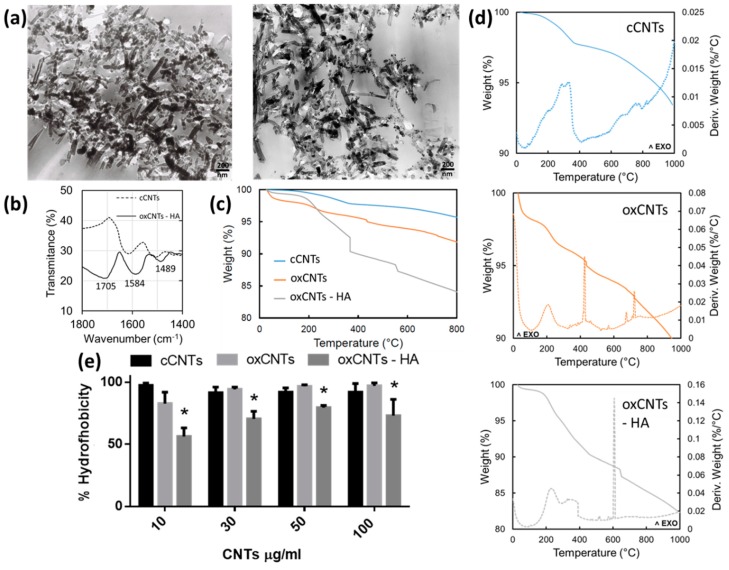
Physical characterization for as prepared nanovectors. In panel (**a**) is shown the electron micrographs cCNTs (**left**) and oxCNT’s-HA-CPT (right). FTIR spectra of cCNTs (dotted line) and oxCNTs-HA (solid line) showing the most representative bands in the range from 1800 to 1400 cm^−1^ (**b**). Thermogravimetric analysis in conjunction with differential scanning calorimetry curves are showed in figures (**c**,**d**), respectively. Finally, the hydrophobicity index is shown in (**e**) (*Bars* indicate the mean ± SD, *n* = 5, **p*< 0.05).

**Figure 2 nanomaterials-09-01572-f002:**
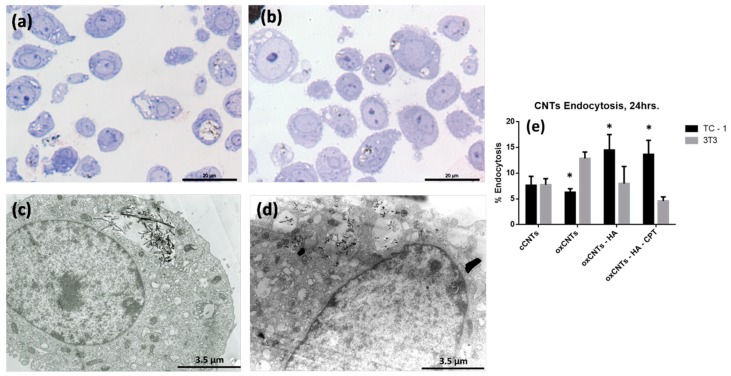
Nanovectors cell uptake. (**a**) Control TC–1 Cells (cCNTs). (**b**) Treated TC–1 Cells with oxCNTs-HA-CPT. (**c**) Electron micrograph of a TC–1 cell with endoplasmic vesicle with cCNTs. (**d**) Electron micrograph of a TC–1 cell with multiple endoplasmic vesicles with oxCNTs-HA-CPT. (**e**) Quantization of cells with CNTs endocytic vesicles (*Bars* indicate the mean ± SD, *n* = 5, **p*< 0.05).

**Figure 3 nanomaterials-09-01572-f003:**
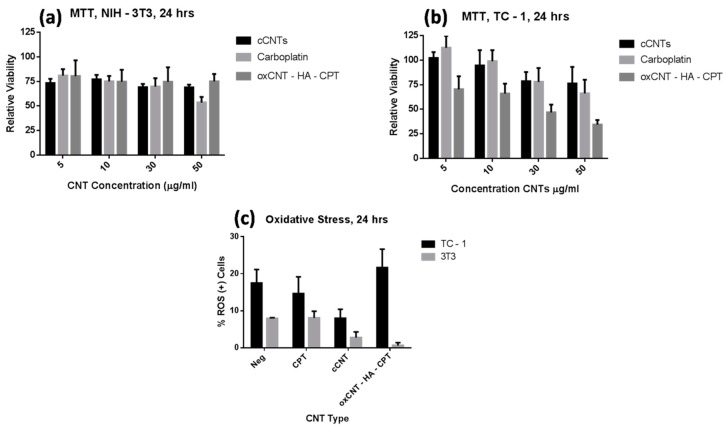
Cytotoxicity and oxidative stress induced by nanovectors. (**a**) 24 h MTT cytotoxicity assay of NIH/3T3 cells. (**b**) 24 h MTT cytotoxicity assay of TC-1 cells. (**c**) 24 h DHE ROS determination assay. (*Bars* indicate the mean ± SD, *n* = 7, **p*< 0.05).

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
