# Peer review of "Hyaluronate Functionalized Multi-Wall Carbon Nanotubes Filled with Carboplatin as a Novel Drug Nanocarrier against Murine Lung Cancer Cells"

_nanomaterials, 2019, doi:10.3390/nano9111572_

Round 1
Reviewer 1 Report
Authors present here the importance of a multi-wall carbon nanotube (CNTs) as a carrier of the drug carboplatin (CPT) into the lung cancer cells. The main goal is to develop a nanovesicle that can be mainly recognized by cancer cells that can receive them and internalize them by endocytosis. Considering the increased amount of the cellular membrane hyaluronan (HA) receptor CD44, authors prepared HA-CNT that in turn may carry CPT. This is a great idea, considering the low efficacy of the drugs alone, the cytotoxicity of normal cells which result to unpleasant side effects to a patient. There are, however, some points to consider.
Major considerations:
1) Authors have used the MTT method to show the cytotoxicity of CNTs and carboplatin. The experiment was performed in 24h. What is the effect of a longer treatment, such as 48h, on both normal and cancer fibroblasts? Authors should make a control of different timepoints to show the efficacy of the treatment of cells with at least the oxCNTs-HA-CPT in the concentration they decided to use, i.e. 10ug/mL.
2) In Fig3c, how do authors explain the decrease of %ROS in normal cells?
3) They are missing the p- values of the significances of the statistical analyses in both figures and legends.
4) MTT assay is an important experiment to show viability and proliferation of cells. Authors have declared at the end of the discussions that there are future experiments to perform to examine the mechanism by which this drug carrier acts. However, it would be important to show here a confirmation of this decrease of viability in cancer cells or by examining the cell apoptosis or at least by showing differences in cell morphology.
Minor considerations:
1) How authors prepared the solutions of all CNTs for the cells treatments? Did you use aqueous solution or dissolved directly in the culture medium? Does the culture medium contain FBS during the adhesion and treatment?
2) Please, check some typing errors in English: line 138 (add the subject), line 169 (CI instead of IC), line 172 “analyses”, Fig3c “Hydrophobicity”, line 295 “[35]. In”, line 300 check the all phrase.
Author Response
Thank you very much for your comments, we really appreciate your opinion regarding our study. In the corrected manuscript we considered all your corrections and suggestions.
Major considerations:
1) Authors have used the MTT method to show the cytotoxicity of CNTs and carboplatin. The experiment was performed in 24h. What is the effect of a longer treatment, such as 48h, on both normal and cancer fibroblasts? Authors should make a control of different timepoints to show the efficacy of the treatment of cells with at least the oxCNTs-HA-CPT in the concentration they decided to use, i.e. 10ug/mL.
A: Regarding this point, in the first instance, the decision to take only the 24-hour cytotoxicity point was made based on a previously published study, where the kinetic of the MWCNTs loaded with platinum-based drugs administered intravenously in mice is evaluated, and it was observed that the maximum peak of residence in the lung is at 24 hours, and from there begins to be redistributed to the organism of metabolism. I enclose the paper reference.
Li, J.; Pant, A.; Chin, C.F.; Ang, W.H.; Ménard-Moyon, C.; Nayak, T.R.; Gibson, D.; Ramaprabhu, S.; Panczyk, T.; Bianco, A.; et al. In vivo biodistribution of platinum-based drugs encapsulated into multi-walled carbon nanotubes. Nanomedicine Nanotechnology, Biol. Med. 2014, 10, 1465–1475.
Also, we are performing an experiment to obtain cytotoxicity at 12 and 48 hours, but this experiment is not already finished yet, so it was not added to the new version of the manuscript.
2) In Fig3c, how do authors explain the decrease of %ROS in normal cells?
A: We have found some papers that support that CNTs possess electron receptor properties inherent in their molecular structure, which makes them good free radical scavengers and have been added to the manuscript in the discussion section (line 342):
"On the other hand, we observe that the ROS levels of the fibroblasts were reduced when treated with the CNTs, this may be possible because the CNTs have a great electronic affinity on their molecular orbitals of the carbon atoms of their graphene walls, acting as a free radical scavenger [44,45], in addition, we observe that this effect is enhanced by the addition of HA since it has antioxidant activity as well.[46]"
Galano, A. Carbon nanotubes: Promising agents against free radicals. Nanoscale 2010, 2, 373–380.
Fenoglio, I.; Tomatis, M.; Lison, D.; Muller, J.; Fonseca, A.; Nagy, J.B.; Fubini, B. Reactivity of carbon nanotubes: Free radical generation or scavenging activity? Free Radic. Biol. Med. 2006, 40, 1227–1233.
Ke, C.; Sun, L.; Qiao, D.; Wang, D.; Zeng, X. Antioxidant acitivity of low molecular weight hyaluronic acid. Food Chem. Toxicol. 2011, 49, 2670–2675.
3) They are missing the p- values of the significances of the statistical analyses in both figures and legends.
A: Referee is right, we apologize for these omissions.
4) MTT assay is an important experiment to show viability and proliferation of cells. Authors have declared at the end of the discussions that there are future experiments to perform to examine the mechanism by which this drug carrier acts. However, it would be important to show here a confirmation of this decrease of viability in cancer cells or by examining the cell apoptosis or at least by showing differences in cell morphology.
A: I greatly appreciate your advice and we have taken it into account, we are conducting the necessary experiments to evaluate cell morphology by investigating signs of apoptosis in cells, however, as in point 1, we have not included the results due to time issues to send our reply.
Minor considerations:
1) How authors prepared the solutions of all CNTs for the cells treatments? Did you use aqueous solution or dissolved directly in the culture medium? Does the culture medium contain FBS during the adhesion and treatment?
A: Referee is right, CNTs were dissolved in culture medium that contain FBS. The culture medium contain FBS during the adhesion and treatment and this have been added to the manuscript in the methods section (line 130):
"For the cell capture, viability and ROS assays, the cells were treated with the different types of CNTs, which were resuspended in the corresponding culture media for each cell line (supplemented with FBS)."
2) Please, check some typing errors in English: line 138 (add the subject), line 169 (CI instead of IC), line 172 “analyses”, Fig3c “Hydrophobicity”, line 295 “[35]. In”, line 300 check the all phrase.
A: Referee is right, we apologize for these mistakes and have been corrected them in the manuscript.
Reviewer 2 Report
The Authors report the synthesis of hyaluronic acid functionalized multi-walled carbon nanotubes as carrier for the delivery of carboplatin to cancer cell. Even if the issues treated in this paper could have relevance in nanomedicine field, I believe that the paper can’t be accepted for publication in this Journal. The main criticism is represented by the lack in novelty, since many papers have long reported the functionalization of carbon nanotubes with hyaluronic acid and also with other biopolymers for the targeted delivery of toxic drugs to cancer cells (see as example, S. K. Prajapati et al., Hyaluronic acid conjugated multi-walled carbon nanotubes for colon cancer targeting, Int. J. Biol. Macromol., 2019, 123, 691-703). Moreover, the coupling of hyaluronic acid with the carboxyl functionalities of carbon nanotubes is not clear; the coupling agents EDC and NHS are normally used for the formation of an amide. Do the Authors report a covalent bond between two carboxylic functionalities? This issue must be addressed before the paper could be considered for publication in this or in other Journals.
Author Response
Thank you very much for your comments, we really appreciate your opinion regarding our study. In the corrected manuscript we considered all your corrections and suggestions.
1.- The main criticism is represented by the lack in novelty, since many papers have long reported the functionalization of carbon nanotubes with hyaluronic acid and also with other biopolymers for the targeted delivery of toxic drugs to cancer cells (see as example, S. K. Prajapati et al., Hyaluronic acid conjugated multi-walled carbon nanotubes for colon cancer targeting, Int. J. Biol. Macromol., 2019, 123, 691-703).
A: Referee is right, there are numerous papers that describe the functionalization of carbon nanotubes with hyaluronic acid and other biopolymers for the targeted delivery of toxic drugs to cancer cells, however, this is the first report of HA functionalized multi-wall carbon nanotubes filled with carboplatin as a novel drug nanocarrier against murine lung cancer cells. We appreciate that you share with us the study of S. K. Prajapati et al., Hyaluronic acid conjugated multi-walled carbon nanotubes for colon cancer targeting, Int. J. Biol. Macromol., 2019, 123, 691-703), this reference and its results were included in our corrected manuscript on the introduction section (line 45, Ref 13):
"This compound act as a ligand of the CD44 receptor, which is overexpressed in a large number of tumors and has been widely used to biodirect MWCNTs for drug delivery."[11–13]
Moreover, the coupling of hyaluronic acid with the carboxyl functionalities of carbon nanotubes is not clear; the coupling agents EDC and NHS are normally used for the formation of an amide. Do the Authors report a covalent bond between two carboxylic functionalities? This issue must be addressed before the paper could be considered for publication in this or in other Journals.
A: Regarding this point, I sincerely apologize. An error has occurred in the writing, we have omitted the use of polyethyleneimine (PEI) in the manuscript, which serves as a coupler for carrying out the amidation reaction with hyaluronic acid. This error has been corrected in the manuscript in the methods section (line 83):
"...and the solution of polyethyleneimine (20 mg in 2 ml of DMSO) as coupler is added, allowing to react for 48 hrs, then separated by centrifugation and left in the oven at 60 ° C until dry."
Additionally, I attach evidence of the binding of HA to the CNTs of the FTIR spectrum, where the correct functionalization of this compound can be observed. Thank you very much for your comments.

Reviewer 3 Report
This is an interesting work on the use of multi-walled carbon nanotubes functionalized using hyaluronate and filled with carboplatin, for lung cancer therapy. The manuscript is well written and can be considered for publication once the following comments are addressed:
Line 178: This should refer to Figure 1'a'.
2. Line 209 mentions that Figure 2a shows the uptake of cCNTs by both TC-1 and NIH/3T3. However the figure legend for Figure 2a says that it is Control TC–1 Cells (cCNT's). Please fix this.
3. The statistical significance between groups must be shown for Figure 3.
4. Please state what the negative control is in Figure 3c.
5. Lines 223-224 - There is something missing in this sentence: 'TC-1 tumor cell line, these cells mouse pulmonary epithelium transformed with'
6. What was the loading efficiency of Carboplatin on these CNTs? No studies have been done to determine the loading and retention of carboplatin within the CNTs.
7. For lung cancer studies, A549 is most the commonly used cell line, since it is a human lung adenocarcinoma cell line. Can the authors elaborate more on why TC-1 murine lung cancer cell line was used for their experiments?
8. The authors should include a conclusions section to summarize their findings and future planned experiments.
Author Response
Thank you very much for your comments, we really appreciate your opinion regarding our study. In the corrected manuscript we considered all your corrections and suggestions.
Line 178: This should refer to Figure 1'a'.
A: Referee is right, we have corrected this observation.
Line 209 mentions that Figure 2a shows the uptake of cCNTs by both TC-1 and NIH/3T3. However, the figure legend for Figure 2a says that it is Control TC–1 Cells (cCNT's). Please fix this.
A: Thanks for the comments, we have made the appropriate changes (line 236):
"The uptake of cCNTs (Figure 2a) in TC - 1 cells was low and as expected, oxCNTs - HA - CPT were taken up in a higher amount because of the high expression of the receptor for HA (Figure 2b)."
The statistical significance between groups must be shown for Figure 3.
A: Referee is right, we apologize for these mistakes, which have been corrected.
Please state what the negative control is in Figure 3c.
A: Thanks for the important observation, in the corrected manuscript on the methods section, we have added that in the negative control for ROS, only the culture medium was renewed in the cells, so this variable represents the basal oxidative stress (line 167).
"...for the negative control (basal oxidative stress), the cells had only a renewal of fresh medium supplemented when administered treatments."
Lines 223-224 - There is something missing in this sentence: 'TC-1 tumor cell line, these cells mouse pulmonary epithelium transformed with'
A: The reviewer is correct, the sentence was not well understood as it was written, we have reformulated it to give it more clarity (line 322).
"For this purpose, we use the TC-1 tumor cell line, these cells are mouse pulmonary epithelial cells transformed with HPV-16 E6 and E7 proteins, and form metastasic tumors specifically in lungs when inoculated intravenously[35]".
What was the loading efficiency of Carboplatin on these CNTs? No studies have been done to determine the loading and retention of carboplatin within the CNTs.
A: I appreciate your comment, we are currently working on a method to establish quantification to determine the amount of carboplatin within the CNTs for future publications. Thus, in this work, we have added this issue in the discussion of the present work, as well as that it is being considered as a perspective.
For lung cancer studies, A549 is most the commonly used cell line, since it is a human lung adenocarcinoma cell line. Can the authors elaborate more on why TC-1 murine lung cancer cell line was used for their experiments?
A: Thanks for the observation, we have added to the discussion that the use of TC-1 and 3T3 cells, which are murine cell lines, was mainly because in this line of work, we want to perfect the formulation and characteristics of the nanovector, to test it in a near future in an animal model in vivo. In this sense, the A549 cells will also be evaluated once we have optimized the nanovector. (line 318):
"Although the A549 cell line (human pulmonary adenocarcinoma) is widely used to test the cytotoxicity and safety of nanomaterials targeting the lung, we decided to apply our nanovectors first in an animal tumor model because we intend to perfect the ideal composition of the nanovector to scale it in an animal in vivo in the near future."
The authors should include a conclusions section to summarize their findings and future planned experiments.
A: I really appreciate the advice and we have considered it, we added the conclusions section and I feel that it was an excellent contribution to the paper, since it helps to establish the main idea and the results, in addition, it helps to propose future experiments to improve our nanovector (line 352):
"In this research we have demonstrated the therapeutic potential of a nanovector based on MWCNTs in vitro. The physicochemical properties such as the functionalization capacity presented by CNTs represent a very useful tool to biodirect them towards tumors, reducing cell death in adjacent healthy tissue. For this purpose, our nanovector was coupled with HA, and although there are many studies that apply this molecule to biodirect nanoparticles, there are no reports in the present literature of the use of CNTs with HA to carry platinum drugs. In addition, CPT, being an FDA-approved drug and widely used in chemotherapy for lung cancer, becomes a viable option to optimize its delivery. The results of our work indicate that the nanovector is easy to obtain, has adequate functionalization, good hydrophilicity, disperses well in culture medium and preferably attacks tumor cells and that they are internalized in a greater proportion in these, compared to normal cells. Although our results are promising so far, studies are needed to complete toxicity and safety profiles such as determining the amount of CPT loaded in nanotubes, long-term toxicity tests and the in vivo tumor model. These studies will help us to optimize the formulation and functionalization and increase the specific toxicity, the load of the drug and its cellular uptake."
Round 2
Reviewer 1 Report
Authors made the corrections requested by authors, even though some experiments were not concluded because of the strictly short time to give a feed-back to the Editor.
This manuscript, however, remains very interesting.
Author Response
Thank you for your suggestion.
Reviewer 2 Report
The Authors have revised the manuscript according to the suggestions of the reviewers. I believe that the manuscript can be accepted for publication in this Journal after minor revisions.
In particular, the discussion on the results of FTIR experiments must be improved, reporting the evidence of the effective coupling between hyaluronic acid and polyethyleneimine which could be confirmed by the presence of an amide bond between the polymers. Also Figure 1b should be better reported.
